# The Sensor Hub for Detecting the Developmental Characteristics in Reading in Children on a White vs. Colored Background/Colored Overlays

**DOI:** 10.3390/s21020406

**Published:** 2021-01-08

**Authors:** Tamara Jakovljević, Milica M. Janković, Andrej M. Savić, Ivan Soldatović, Petar Todorović, Tadeja Jere Jakulin, Gregor Papa, Vanja Ković

**Affiliations:** 1Jožef Stefan International Postgraduate School, 1000 Ljubljana, Slovenia; 2School of Electrical Engineering, University of Belgrade, 11000 Belgrade, Serbia; piperski@etf.rs (M.M.J.); andrej_savic@etf.rs (A.M.S.); 3Institute of Medical Statistics and Informatics, Faculty of Medicine, University of Belgrade,11000 Belgrade, Serbia; ivan.soldatovic@med.bg.ac.rs; 4Faculty of Engineering, University of Kragujevac, 34000 Kragujevac, Serbia; petar@kg.ac.rs; 5FTŠ Turistica, University of Primorska, 6320 Portorož, Slovenia; tadeja.jerejakulin@upr.si; 6Computer Systems Department, Jožef Stefan Institute, 1000 Ljubljana, Slovenia; gregor.papa@ijs.si; 7Laboratory for Neurocognition and Applied Cognition, Faculty of philosophy, University of Belgrade, 11000 Belgrade, Serbia; vanja.kovic@f.bg.ac.rs

**Keywords:** sensor hub, reading in children, developmental differences, background colors, overlay colors

## Abstract

This study investigated the influence of white vs. 12 background and overlay colors on the reading process in twenty-four school-age children. Previous research reported that colors could affect reading skills as an important factor in the emotional and physiological state of the body. The aim of the study was to assess developmental differences between second and third grade students of an elementary school, and to evaluate differences in electroencephalography (EEG), ocular, electrodermal activities (EDA) and heart rate variability (HRV). Our findings showed a decreasing trend with age regarding EEG power bands (Alpha, Beta, Delta, Theta) and lower scores of reading duration and eye-tracking measures in younger children compared to older children. As shown in the results, HRV parameters showed higher scores in 12 background and overlay colors among second than third grade students, which is linearly correlated to the level of stress and is readable from EDA measures as well. Our study showed the calming effect on second graders of turquoise and blue background colors. Considering other colors separately for each parameter, we assumed that there are no systematic differences in reading duration, EEG power band, eye-tracking and EDA measures.

## 1. Introduction

Learning to read is a complex process involving both perception and cognition, via the integration of visual and auditory information processing and memory, attention and language skills [1]. Therefore, reading is a taught skill depending upon a range of perceptual processes and cognitive abilities affecting learning over time and across development [2,3,4]. One of the key objectives of early education is the learning of reading, and a number of research studies have explored the process of reading skills acquisition in children [2,5,6,7]. Depending on their individual set of underlying abilities, children will have different developmental profiles of reading skill obtainment, while weaknesses in some abilities may cause reading impairments over time. Individual differences in the learning of reading may originate from biological and environmental factors, shaping the development of the brain systems involved in the reading process [8]. The current knowledge emphasizes the importance of identifying and treating reading difficulties as early as possible, since they may impair academic achievements and increase the risk of social, emotional and mental health problems in children. More specifically, poor reading skills are shown to be associated with increased risks for school dropout, attempted suicide, incarceration, anxiety, depression, and low self-concept [9]. 

There is some limited evidence that colors may impact the reading process, specifically with early school-age children, and those with reading disabilities [10,11]. Going back a few decades [12], it has been shown that the role of colors in reading dates back to 1958. Jansky [13] reported the case of a student with a reading deficit who was able to recognize words printed on yellow paper, but unable to recognize words printed on white paper. Previous studies considered the influence of background, text or overlay color on the actual reading process in children [12,14,15]. While more recent studies have shown that colors do not influence the reading process and that this could be a placebo [16], others have found that colors may be particularly effective for early readers in school-age children [17]. 

As reading involves sensory integration, attention and memory, those processes may be reflected in the psycho-physiological states of the individual engaged in the reading task. Those states are a result of underlying neural and physiological processes, which are measurable and quantifiable by different biosignal modalities. The goal of this study was to employ multimodal sensor measurements to examine the influence of the color of the content on the reading task for children at different developmental stages. More specifically, we have employed measurements of electroencephalography (EEG), eye-tracking, electrodermal activity (EDA) and heart rate variability (HRV) to assess the influence of background and overlay color on reading performance in second and third grade students at elementary school. We aimed to address the mechanisms of color’s influence on the reading process through electrophysiological correlates of the reader’s state, while taking into account the developmental aspect of reading acquisition. 

This study sheds light on the underlying neural, physiological and behavioral processes accompanying the reading task in children. Moreover, this is an initial step towards the possibility of including color into reading content to improve reading skills in children at different developmental stages, and addressing the question, “Why would the effect of text background color (if there is one) vary according to age?” One prediction could be that the younger children would have more difficulties reading the text on the intense color background in comparison to a pastel background. This is because the reading process is still not automated, and so it makes it harder for them to read in such a context in comparison to the older children who are proficient in reading [18,19,20]. An alternative hypothesis would be that the older children would struggle more with any color except for white, because of the experience they have reading on white as a default background in comparison to any other background that is novel (and distracting) for them [21,22,23].

## 2. Materials and Methods

### 2.1. Participants

Twenty-five healthy participants were randomly chosen from eight classes of the second and third grades of the elementary school “Drinka Pavlović” in Belgrade, respectively. These two age groups were selected intentionally, because children in Serbia start to learn letters in the first grade and the biggest shift in the reading happens between the second and third graders. The reading experience was compatible between the two grade groups, as both of these grades spend equal amounts of time in school (8am–3.20pm) and they finish all their homework at school as well. The participants’ (10 boys and 15 girls) ages ranged from 8 to 9 years. The inclusion criteria were that children have no reading and learning disabilities or attention disorders, and have normal or corrected to normal vision according to teachers assessment. Only one child was excluded from the analysis due to the large artefacts in the acquired signals and his data were not used in the statistical analysis.

All the subjects underwent the same experimental conditions and participated individually in the small school classroom during the regular time period of the daily school classes. Each child received a short instruction about the experiment setup and design. Consent forms were provided through the school director for each child who participated in the study. After the experiment, every child received a small present and diploma for participation in the experiment process.

The ethical committee of the Psychology Department of the University of Niš approved the experimental procedure.

### 2.2. Experiment Setup

During the experiment, each participant was sitting on a chair at a table in the front of a computer monitor and keyboard. At the beginning of the experiment, participants got the instruction to read quietly for themselves the text from the stimuli presentation shown on the computer monitor, and to press the space button for the next slide of stimuli presentation. The experiment was run applying the pseudo randomization of color background/overlay order, starting always with a referent slide (black text on white background). No other color was fixed/related to a certain text. Therefore, in this way, any other factors apart from the actual color would be averaged out (paragraph complexity such as vocabulary, syntax, etc., as well as semantic/affective content). Details about the “experiment” content are explained in the Section Experiment Design.

During the reading process, physiological data were acquired using a sensor hub composed of an eye-tracking system and a portable multimodal EEG/ECG/EDA system, Figure 1. Two laptops were used for real-time data monitoring and storage: one laptop for eye movement monitoring (with additional external computer monitor and keyboard in front of the participant) and another for EEG/ECG/EDA monitoring.

An SMI RED-m 120 Hz portable remote eye tracker (iMotions, Copenhagen, Denmark) was placed below the computer monitor in front of the participant, and it was fixed in place to keep it from accidentally moving. An adjustable chinrest was used to ensure the same distance from the monitor and table for each participant (the chinrest was 57 cm away from the eye-tracking sensor and 16 cm above the table) [24]. The SMI software was used for stimuli presentation (Experiment Centre 3.7) and data collection (iView RED-m).

The EEG and ECG signals were recorded using a mobile 24-channel EEG amplifier (SMARTING, mBrainTrain, Belgrade, Serbia) wirelessly communicating with a laptop via Bluetooth. Twenty-two monopolar EEG channels of Greentek Gelfree-S3 cap (10/20 locations: Fp1, Fp2, F3, F4, C3, C4, P3, P4, O1, O2, F7, F8, T7, T8, P7, P8, Fz, Cz, Pz, AFz, CPz, POz) were recorded. The ground was located at FPz and FCz was used as the reference site. One channel of the amplifier was connected to the surface SKINTACT ECG electrode placed in the left chest region, over the heart, to record ECG signal as a reference for heartbeat detection. EEG and ECG signals were acquired with 24-bit resolution and 250 Hz sampling rate. The skin–electrode impedances were below the manufacturer recommended value of 1 kOhm, prior to the tests.

One channel of the amplifier was used for the synchronization of electrophysiological recordings and eye tracking data. A small photosensitive sensor registering the changes on the screen after each slide was used with the changes of the black and white screen (200 ms each), and it sent a trigger for the synchronization of the multimodal EEG/ECG/EDA system and eye tracking system for each event (slide).

We used a research prototype for galvanic skin response (electrodermal activity, EDA) recording [25] that communicates with a laptop via Bluetooth. The sampling rate for EDA data was 40 Hz. The SMARTING application has Lab Streaming Layer (LSL) compatibility which enabled synchronization between the EDA and EEG/ECG data within a single file of XDF format. 

### 2.3. Experiment Design

#### 2.3.1. Stimuli

Participants read a story on the computer monitor. The story was at an adequate level for the second/third grade of elementary school and was selected from the school’s literature for the Serbian language course. Participants were unfamiliar with the text used in the study.

The story “St Sava and the villager without luck” was split into 13 paragraphs: the 1st slide was a referent one with a white background with black letters, then there were 6 slides with black letters on red, blue, yellow, orange, purple and turquoise backgrounds, and the next 6 slides were presented in the overlay manner which looks like covering black text on a white background with a colored foil (calculated by the algorithm described in the section *Color Calculation*).

The experiment started with calibration and the validation method would stop on the black slide so the researcher had time to launch the multimodal EEG/ECG/EDA system and eye-tracking system for data acquisition. Next, on the researcher’s instruction, the child would press the space button on the keyboard and the first slide with the text would appear on the computer monitor. The participant read text to themselves and then pressed the space button for the next slide to continue the text.

After finishing the test, the researcher posed questions (recommended for exercise in literature after the story) to check whether the children read the text carefully or not.

#### 2.3.2. Color Calculation

All colors (color shades) used for designing the slides (stimuli) were defined within the RGB color model and each individual color was expressed as an RGB triplet ([R,G,B]), where the value of each additive primary color component can vary from 0 to 255. A list of background shades in the slides with colored backgrounds (and black text) with associated numerical values of their RGB triplet is as follows: red (“red”, [255,0,0]), blue (“blue”, [0,0,255]), yellow (“yellow”, [255,255,0]), orange (“orange”, [255,128,0]), purple (“purple”, [255,0,128]) and turquoise (“turquoise”, [0,255,255]. White and black shades were defined by triplets [255,255,255] and [0,0,0], respectively. The RGB components of the background and text in the slides with “overlay effect” were calculated according to the following formula:Overlay Component = Shade Component * Opacity + (1 − Opacity) * Underlay Shade
where Opacity value was set to 0.5, Shade Component was selected from one of the previously listed background color shades, and the Underlay Shade value was 0 for black and in the case of a white background was 255.

The resulting RGB triplets for the shades of the text and background for slides with overlay effect were as follows: overlay red (“red O”, text—[128,0,0], background—[255,128,128]), overlay blue (“blue O”, text—[0,0,128], background—[128,128,255]), overlay yellow (“yellow O”, text—[128,128,0], background—[255,255,128]), overlay orange (“orange O”, text—[128,64,0], background—[255,192,128]), overlay purple (“purple O”, text—[128,0,64], background—[255,128,192]) and overlay turquoise (“turquoise O”, text—[0,128,128], background—[128,255,255]).

#### 2.3.3. Data Processing

Eye tracking data analysis and visualization was performed using SMI BeGaze ^TM^ 3.7 software (SensoMotoric Instruments, Teltow, Germany). The selected eye-tracking parameters were fixation count, fixation frequency (count/second), fixation duration total (ms), fixation duration average (ms), saccade count, saccade frequency (count/second), saccade duration total (ms), and saccade duration average (ms). EEG/ECG/EDA data were analyzed using Matlab ver. 8.5 (Mathworks, Natick, MA, USA) in the manner described below.

For each subject and electrode site, the EEG signal was processed in order to calculate the band power in five predefined frequency bands, as follows: delta (0.5–4 Hz), theta (4–7 Hz), alpha (7–13 Hz), beta (15–40 Hz) and whole range (0.5–40 Hz). The median value of the EEG band power (for each frequency band) was determined for 13 time-epochs coinciding with the reading of the content of each presented slide. The median EEG power in each frequency band was calculated by raw continuous signal band-pass filtering (4th order Butterworth filter with cut-off frequencies defined by individual band’s frequency range), squaring, segmenting into 13 epochs (determined as time intervals between each slide’s onset and offset), and median averaging to a single power value over signal samples of each epoch (i.e., over each slide’s duration). The applied processing resulted in 65 median power values (i.e., 13 slides × 5 frequency bands) for each EEG channel of each subject. Median power calculation was applied since it is less likely to be affected by outliers in the EEG power samples that occur with temporary movement artefacts than the mean power.

The heart activity signal was band-pass filtered using an FIR filter in the range 1–45 Hz (750 points), after which the Pan–Tompkins algorithm [26] for the extraction of heart activity beats was applied. Beat-to-beat intervals (BBI, the time between two successive heart activity peaks) were calculated. Heart rate variability (HRV) parameters [27] were calculated from the BBIs for 13 time-epochs coinciding with the reading of the content of each presented slide (Table 1). The applied processing resulted in 14 HRV parameter values in one subject for each of the 13 slides. 

The mean value of the EDA data was calculated as a representative value of electrodermal activity for 13 time-epochs coinciding with the reading of the content of each presented slide. The applied processing resulted in 13 mean values for each subject (one mean EDA value per slide). 

### 2.4. Statistical Methodology

The results are presented as count (%), means ± standard deviation, or depending on data type and distribution. The groups are compared using parametric tests and independent samples t tests. All *p* values less than 0.05 were considered significant. There were no substantial deviations from the parametric testing assumptions. All data were analyzed using SPSS 20.0 (IBM Corp. Released 2011. IBM SPSS Statistics for Windows, Version 20.0. Armonk, NY: IBM Corp.) and R 3.4.2. (R Core Team (2017). R: A language and environment for statistical computing. R Foundation for Statistical Computing, Vienna, Austria. URL https://www.R-project.org/). It is noteworthy that Bonferroni corrections were applied in all the statistical analyses with multiple comparisons.

## 3. Results

### 3.1. White (Default) Background—Reading Results

Grade comparisons (second vs. third) regarding the examined parameters for white color only are presented in Table 2. A significant difference was obtained regarding EEG frequency bands (Alpha, Beta and Theta) and ECG parameters (SDNN, CVRR and STD HR). In all other parameters, we observed no significant difference between second and third graders. All EEG power bands are higher in the second grade group (except Delta band), compared to third graders. An opposite trend is shown in the SDNN, CVRR and STD HR parameters, whereby third graders achieved higher scores in comparison to second graders.

### 3.2. Background and Overlay Colors—Reading Results

In Table 3 we show the overall results which were calculated by subtracting the parameters acquired for the white color from the parameters acquired for each of the background and overlay colors (normalized values). Second and third graders were then compared on each of the parameters measured in the study, namely, reading duration, EEG, eye tracking, EDA and HRV.

The results demonstrated that students in the second grade differed significantly from the students in the third grade consistently on a few HRV parameters, in particular, SDNN (yellow, turquoise and turquoise O), CVRR (for orange, turquoise and blue O), and STD HR (blue, yellow, turquoise, blue O, yellow O, orange O and turquoise O). Besides these differences, significant differences were found in RMSSD (for turquoise and blue O). In each of these cases second grade students scored higher (or positively) in comparison to the third grade students who scored lower (or negatively), which is marked with an orange color in Table 3. Out of all the parameters in the study, only in the case of saccade duration total for the blue O did the third graders score significantly higher than the second graders, which is marked with a green color in Table 3.

In Figure 2, the normalized SDNN, CVRR and STD HR values across grades and colors are presented.

Given that there were no systematic differences across colors in Table 3 for reading duration, eye-tracking, EEG or EDA parameters, for a subsequent analysis all twelve background and overlay colors were averaged together in order to examine the differences between younger and older children.

Grade comparisons (second vs. third) regarding the examined parameters over the averaged scores for all colors together are presented in Table 4. A significant difference was obtained regarding reading duration, and median Alpha, Betha, Delta and Theta power bands, as well as for the whole range, then for fixation duration total, fixation duration average and EDA. In all other parameters we observed no significant difference between second and third graders.

Regarding reading duration, third graders took a significantly longer time to complete the reading. All EEG power bands (Alpha, Betha, Delta, Theta and whole range) as well as EDA were higher in the second grade group, compared to third graders. An opposite trend was found for the eye-tracking measures, whereby both fixation duration total and fixation duration average were higher in the third graders in comparison to the second graders.

## 4. Discussion

We have evaluated differences in reading duration, EEG, eye tracking, EDA and HRV parameters in 24 children (12 second and 12 third grade students in elementary school) using simultaneously monitored sensor signals. 

The literature regarding EEG power bands (Alpha, Beta, Delta, Theta) has been shown to have a general developmental decreasing trend with increasing age [28,29,30]. However, in the case of a specific mental activity, such as reading in this study, little is yet known about developmental changes in the contribution of EEG power bands [31]. When kids were given a task to read on a white color (which is a standard and everyday background color), they showed a reduction in Alpha, Beta and Theta power bands with age (Table 2). Given that this is a pilot study in this sense, it remains to be resolved whether this is specifically a developmental trend, or one that has to do with experience, or both in combination. To fully resolve this issue, one would also need to have a much wider range of age groups tested. 

A correlation between HRV parameters and age in infants and during childhood, caused by progressive maturation of the autonomous nervous system, is known in the literature [31,32,33,34]. However, the impact of age on HRV parameters is more extensive in infancy and the early childhood period (up to eight years) [31]. In this study, participants were 8 and 9 years old, so they were in years of life when the age impact was much less significant on HRV parameters [36]. Consequently, the level of stress in our study was expected (and found) to be lower in the group with a higher strain [36]. Namely, SDNN, CVRR and STD HR were found to be lower in younger children when reading on a white background (Table 2). However, we would not predominantly attribute these findings to the developmental causal factor, but at least to some extent to experience in reading. 

This study gives further support to the existing findings that colors may play an important role in the reading process [12,14,15,17,37,38,39,40,41]. When normalization on the white background was performed (by subtracting white from each of the other background and overlay colors), systematic differences (on at least two colors) were found regarding HRV parameters (normalized SDNN, CVRR, and STD HR values), with second graders scoring higher on these parameters (Table 3). The corresponding graphics across colors are presented in Figure 2, where it could be observed that blue and turquoise backgrounds have a calming impact (increasing normalized SDNN, CVRR, and STD HR values) on second graders, which is in accordance with the previous reports [18,19,20] and partially in accordance with our first hypothesis. No systematic differences were found across colors between younger and older children in reading duration, EEG power band, eye-tracking and EDA measures. This is why, based on the results presented in Table 3, for the following analyses all twelve background and overlay colors were averaged together, and we examined differences between second and third graders in that case (Table 4). Based on the results presented in Table 4, we observed a significant difference across multimodal sensor measurements in terms of reading on background and overlay colors in contrast to reading on the white background color. 

The findings concerning reading duration on background and colored overlays showed that the third grade students achieve longer reading durations in comparison to the second graders. This could be explained by the fact that third graders or older children need more time to adapt to unexpected text and stimuli such as color [21,22,23], which is in line with our alternative hypothesis. Additionally, previous studies have shown that older children are slower in reading than younger children because they demand longer fixation duration and saccades [41,42,43,44,45,46,47], skipping words more frequently than younger children [48,49]. Concerning the eye-tracking measures, it was in fact found that third graders have longer fixation duration totals and fixation duration averages in comparison to the second-graders, which is a result that is in line with the above-mentioned assertion that older children take longer to read. This indicates the higher mental load in the reading task in older children. 

A similar pattern of EEG results found for the white background was also found for the background and overlay colors. However, the number of parameters that showed a significant difference between second and third graders and significance levels was much more prominent for background and overlay colors in comparison to the white background. The study shows significant differences between the second and third graders in all four EEG power bands (Alpha, Beta, Delta and Theta), as well as in the whole range based on the averaged results from the twelve background and overlay colors. 

Similarly, it is important to mention that EDA linearly correlates to arousal and reflects cognitive activity and emotional response [50,51], and it is the most used psychophysiological measure of arousal [52]. The higher the arousal is, the higher the electrodermal activity is. In the present study, it was found to be higher among second graders, from which we conclude that they had a higher stress level. 

## 5. Conclusions

The aim of this study was to assess developmental differences in second and third grade students in elementary school regarding reading on white vs. colored overlay and background, and to evaluate differences in reading duration, and brain, eye, electrodermal and heart activities. Evaluating all the findings and results derived during reading on the white color and all 12 background and overlay colors, we can conclude that there is a decreasing trend with age regarding EEG power bands (Alpha, Beta, Delta, Theta) that is shown in the comparison between second and third grade students. In addition, second graders show lower scores in the reading duration and eye-tracking measures (fixation duration total and fixation duration average), which confirms the (alternative) hypothesis that older children need more time to adapt to unexpected text, and have longer fixation durations on words during reading. Comparing HRV parameters in second and third graders during reading on the white color, we found lower scores in second graders and higher scores in 12 overlays and background colors compared to third graders, especially for the SDNN, CVRR, and STD HR measures. The highest values of normalized SDNN, CVRR and STD HR among students were reached with the turquoise and blue background colors, which could be the result of a calming effect during task performance caused by the background color. Furthermore, we have found that EDA linearly correlates with level of arousal (tension and stress), whereby we have found higher values in younger children. Across single colors and their influences on measures during the reading task, there are no systematic differences in reading duration, EEG power band, eye-tracking and EDA measures, except for HRV measures, as mentioned above.

Thus, the goal of combining different modalities was to find a more objective approach to understanding the developmental differences in children’s reading, as well as to understand the contribution of different modalities and combinations of modalities to the process of reading text on a white vs. colored overlay and background.

In the following work, it will be necessary to move forward from group studies to individual studies in order to determine and establish the individual optimal parameters, as well as colors, corresponding to individual differences in the reading process.

## Figures and Tables

**Figure 1 sensors-21-00406-f001:**
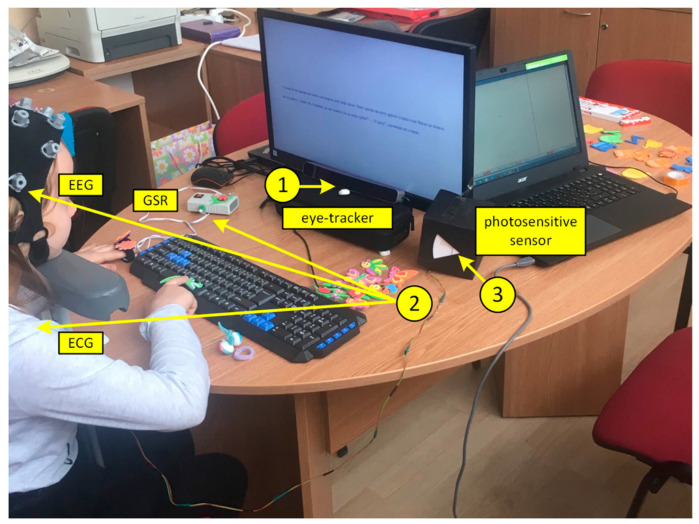
Experiment setup: (**1**) Eye-tracking system; (**2**) Portable multimodal EEG/ECG/EDA system; (**3**) Photosensitive sensor for synchronization of multimodal EEG/ECG/EDA system and eye-tracking system.

**Figure 2 sensors-21-00406-f002:**
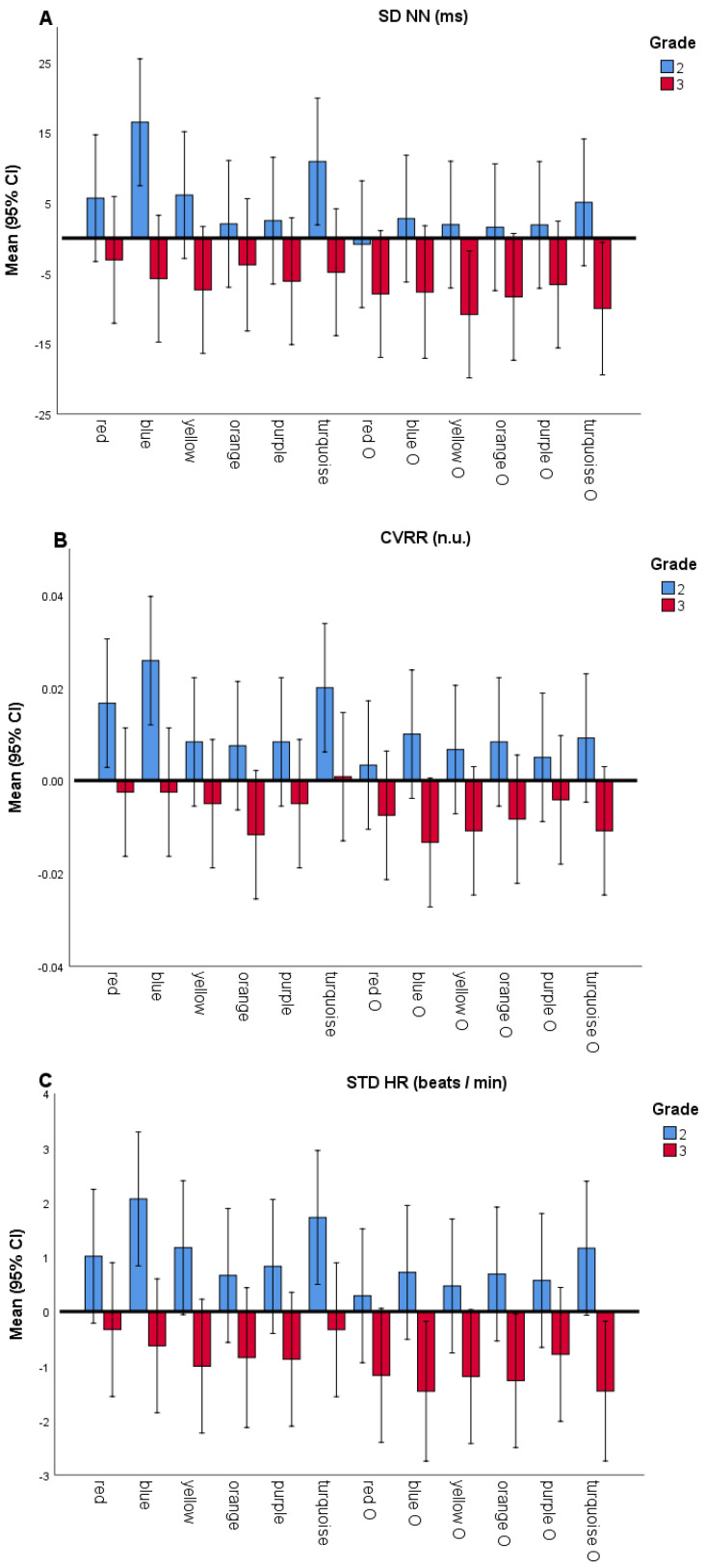
SDNN, CVRR and STD HR by grade and color (normalized on white color). (**A**) Normalized SDNN values (ms), (**B**) Normalized CVRR values (n.u.) and (**C**) Normalized STD HR (beats / min). Normalized values of each parameter are calculated by subtracting the parameter value for white background from the values for each of the background / overlay colors. Bar plots show the normalized data for all background and overlay colors (overlay colors labeled with "o" on x-axis), where second and third grader’s values are presented with blue and red colored bars, respectively. Error bars denote standard errors of the mean.

**Table 1 sensors-21-00406-t001:** Heart rate variability (HRV) parameters.

Parameter	Unit	Description
**Time domain parameters**		
Mean RR	ms	Mean value of BBIs
SDNN	ms	Standard deviation of normal BBIs
Mean HR	beats/min	Mean value of heart rate
STD HR	beats/min	Standard deviation of heart rate
CVRR = SDNN/Mean RR	n.u.	Coefficient of variance of normal BBIs
RMSSD	ms	Root mean square of differences of successive BBIs
NN50	beats	Number of successive BBIs that varied more than 50 ms
pNN50	%	Percentage of successive BBIs that differ more than 50 ms

**Table 2 sensors-21-00406-t002:** Reading duration, EEG, eye tracking, EDA and HRV parameters in second and third grade—significant *p* values are marked as bold.

Parameters	Grade		*p* Value *
	Second (*n* = 12)	Third (*n* = 12)	
Reading duration			
RD (s)	43.38 ± 25.32	49.07 ± 27.63	0.628
EEG parameters (median power band)			
Alpha (μV^2^)	13.19 ± 5.15	5.63 ± 4.28	**0.001**
Beta (μV^2^)	6.05 ± 2.81	3.33 ± 2.40	**0.018**
Delta (μV^2^)	64.45 ± 20.47	71.85 ± 63.54	0.707
Theta (μV^2^)	16.07 ± 7.14	8.35 ± 7.67	**0.018**
Whole Range (μV^2^)	113.3 ± 38.5	97.2 ± 80.5	0.540
Eye tracking parameters			
Fixation Count	38.25 ± 19.62	35.00 ± 9.66	0.620
Fixation Frequency (count/s)	1.01 ± 0.36	0.99 ± 0.67	0.925
Fixation Duration Total (s)	40.87 ± 25.01	46.22 ± 26.32	0.631
Fixation Duration Average (ms)	1088.2 ± 542.9	1331.5 ± 730.9	0.381
Saccade Count	34.92 ± 19.16	28.70 ± 5.23	0.301
Saccade Frequency (count/s)	0.95 ± 0.38	0.86 ± 0.65	0.686
Saccade Duration Total (ms)	741.3 ± 449.4	680.0 ± 276.7	0.711
Saccade Duration Average (ms)	20.93 ± 2.76	23.22 ± 5.61	0.227
EDA value			
EDA (μS)	9.41 ± 3.49	6.20 ± 4.38	0.060
HRV parameters			
Mean RR (ms)	637.4 ± 59.8	680.5 ± 114.4	0.264
SDNN (ms)	35.14 ± 11.95	66.11 ± 46.15	**0.043**
CVRR (n.u.)	0.06 ± 0.02	0.10 ± 0.05	**0.021**
Mean HR (beats/min)	94.87 ± 8.69	90.26 ± 13.75	0.337
STD HR (beats/min)	5.12 ± 1.54	8.40 ± 3.85	**0.016**
RMSSD (ms)	42.50 ± 17.42	84.94 ± 74.74	0.079
NN50 (beats)	12.42 ± 9.44	19.67 ± 16.43	0.298
pNN50 (%)	23.40 ± 18.34	36.30 ± 25.88	0.236

* independent sample t test.

**Table 3 sensors-21-00406-t003:** Differences between second and third graders on reading duration, EEG, eye tracking, EDA and HRV parameters (normalized on white color), (*p* < 0.05 is marked with pale orange color, and *p* < 0.01 is marked with dark orange color). The values in the table represent Cohen’s D effect sizes.

Parameters	Normalized Values											
	Red	Blue	Yellow	Orange	Purple	Turquoise	Red O	Blue O	Yellow O	Orange O	Purple O	Turquoise O
Reading duration												
RD (s)												
EEG parameters (median power band)												
Alpha (μV^2^)												
Beta (μV^2^)												
Delta (μV^2^)												
Theta (μV^2^)												
Whole Range (μV^2^)												
Eye tracking parameters												
Fixation Count												
Fixation Frequency (count/s)												
Fixation Duration Total (s)												
Fixation Duration Average (ms)												
Saccade Count												
Saccade Frequency (count/s)												
Saccade Duration Total (ms)								0.9				
Saccade Duration Average (ms)												
EDA value												
EDA (μS)												
HRV parameters												
Mean RR (ms)												
SDNN (ms)			0.6			0.5						0.2
CVRR				0.8		0.6		0.6				
Mean HR (beats/min)												
STD HR (beats/min)		0.2	0.4			0.5		0.2	0.7	0.5		0.3
RMSSD (ms)						0.5		0.3				
NN50 (beats)												
pNN50 (%)												

**Table 4 sensors-21-00406-t004:** Reading duration, EEG, eye tracking and EDA parameters in second and third grade across all colors together—significant *p* values are marked in bold.

Parameters	Grade		*p* Value *
	Second (*n* = 12)	Third (*n* = 12)	
Reading duration			
RD (s)	42.74 ± 31.81	50.33 ± 27.40	**0.001**
EEG parameters (median power band)			
Alpha (μV^2^)	13.50 ± 5.28	4.67 ± 3.20	**0.001**
Beta (μV^2^)	6.36 ± 2.90	3.71 ± 4.70	**0.001**
Delta (μV^2^)	71.49 ± 37.33	51.23± 48.90	**0.001**
Theta (μV^2^)	17.55± 7.88	6.38± 4.75	**0.001**
Whole Range (μV^2^)	123.36 ± 54.24	74.45± 61.30	**0.001**
Eye tracking parameters			
Fixation Count	35.14 ± 13.97	36.35± 10.31	0.412
Fixation Frequency (count/s)	1.10 ± 0.89	1.01 ± 0.75	0.372
Fixation Duration Total (s)	40.01 ± 30.71	47.47 ± 25.63	**0.028**
Fixation Duration Average (ms)	1038.49 ± 593.15	1283.29 ± 584.55	**0.001**
Saccade Count	31.31 ± 12.36	30.93 ± 8.05	0.753
Saccade Frequency (count/s)	0.99 ± 1.00	0.89 ± 0.71	0.376
Saccade Duration Total (ms)	662.9 ± 293.4	712.0 ± 273.6	0.147
Saccade Duration Average (ms)	21.65 ± 6.09	22.69 ± 3.88	0.093
EDA value			
EDA (μS)	10.13± 3.25	6.32 ± 3.95	**0.001**

* independent sample t test.

## Data Availability

Data of this research is available upon request via corresponding author.

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
