# Peer review of "The Sensor Hub for Detecting the Developmental Characteristics in Reading in Children on a White vs. Colored Background/Colored Overlays"

_sensors, 2021, doi:10.3390/s21020406_

Round 1

Reviewer 1 Report

The authors aimed to investigate the influence of white versus other overlay colours on the reading process in 25 children (8-9 years old).

The topic is novel and of interest, the methodology is good and well reported. 

Author Response

Thank you for your support.

Reviewer 2 Report

Comments and Suggestions for Authors

The manuscript considered the effect of background and overlay colors on the reading process in 2nd and 3rd grade children. Though I don’t have a major concern, I have made several comments given below.  The manuscript offers

Detailed comments

- In abstract: need to report the number of participants
- Page 1, Line 33: It is unclear what "the existing study" is referred to.
- Page 2, Lines 81–88: Two hypotheses were provided, yet no background literature was provided to support the hypotheses.
- Page 2, Section 2.1: It is a small matter, but please indicate whether or not participants gave consent to the study or not (likely via their parents).
- Page 2, Line 91: Provide the gender breakdown.
- Page 3, Line 94: Please clarify if the authors assessed learning disabilities, attention disorders, and eye site using relevant instruments, or based on self-report and/or subjective assessments of the author(s).
- Page 3, Line 116: data real-time monitoring >> real-time data monitoring
- Page 3: I don't seem to find any information about participants' baseline reading level and their reading experience (e.g., how many hours they typically read per week, etc.).
- Page 3, Lines 162-163: It is unclear what happened after the investigator(s) assessed comprehension. Did the investigators record scores or summative evaluation?
- Page 4, Line 187: BeGaze 3.7 software. Like it was done for Matlab in a later sentence, need to provide the company name, and location information. Or, if BeGaze is part of the eye tracking system, clearly indicate so.
- Page 5, Lines 224-235: I am not sure why this is here. It sounds like the author's instruction from a specific journal.

- Methods: given the number of outcome measures, please report how much they are correlated each other to help the reader understand how much distinct information each outcome measure offers.
- Page 6, Section 2.4: Please indicate whether or not there was a substantial departure from parametric test assumptions for any of dependent variables. It is unclear.
- Page 6, Table 2: EEG parameters. Please include the unit (Hz?). And n = 12 for the both grades. Previously, it was noted that one participant's data was excluded.
- Page 7, Table 3: I don't think there is much value in noting that p<0.05 and p<0.01. Nowadays it is more encouraged to report the effect size. So, I also encourage the authors to remove such a distinction based on p values, and include the effect size. And n = 156. It is unclear where that came from.
- Page 9, Figure 2: The figures are not easy to read (e.g., label font size is too small, image quality is low, etc.). In addition, the variable in the horizontal axis is not a continuous variable, so a bar graph is more proper. Or, remove the line that connects the values of an outcome measure.
- Page 10, Line 297: In the methods, it was noted that one participant data was excluded.

- Discussion: Given the two hypotheses were provided, please more clearly indicate whether or not the hypotheses were accepted or failed to be accepted.

Author Response

Point 1:

- In abstract: need to report the number of participants

Response 1: We have now reported the number of participants in the abstract, by rephrasing the following sentence: “The study investigated the influence of white vs 12 background and overlay colours on the reading process in school age children.”  so now it reads: “The study investigated the influence of white vs 12 background and overlay colours on the reading process in twenty-four school age children.” (LINE 24)

Point 2:

- Page 1, Line 33: It is unclear what "the existing study" is referred to.

Response 2: It refers to a current, presented study. We have changed "The existing study" to "Our study". (LINE 33)

Point 3:

- Page 2, Lines 81–88: Two hypotheses were provided, yet no background literature was provided to support the hypotheses.

Response 3: This has now changed; we have added the supporting references in both cases (we have moved references [37-42] that were supported the discussion to [19-24] to support hypothesis as well and we have reordered all references):

“One prediction could be that the younger kids would have more difficulties reading the text on the intense colour background in comparison to pastel background. This is because the reading process is still not automated, and so it makes it harder for them to read in such context in comparison to the older kids who are proficient in reading [19,20,21]. An alternative hypothesis would be that the older kids would struggle more with any colour except for white, because of the experience they have reading on white as a default background in comparison to any other background which is novel (and distracting) for them [22,23,24].” (LINE 81-87)

Point 4:

- Page 2, Section 2.1: It is a small matter, but please indicate whether or not participants gave consent to the study or not (likely via their parents).

Response 4: Yes, this was organised through a school director. Parents of the children who took part in the study were given a consent form prior to the participation in the study.  

Before the sentence: “After the experiment, every child received a small present and diploma for participation in the experiment process.”, we included the following sentence: “Consent forms were provided through the school director for each child who participated in the study.” (LINE 103 to 104)

Point 5:

- Page 2, Line 91: Provide the gender breakdown.

Response 5: We have now specified it by changing the sentence: “ The participants ages ranged from 8 to 9 years” to  “The participants (10 boys and 15 girls) ages ranged from 8 to 9 years.” (LINE 97)

Point 6:

- Page 3, Line 94: Please clarify if the authors assessed learning disabilities, attention disorders, and eye site using relevant instruments, or based on self-report and/or subjective assessments of the author(s).

Response 6: The assessment was made by school teachers. We specified this in the paper by changing the sentence: “Inclusion criteria were that children have no reading and learning disabilities, attention disorders, have normal or corrected to normal vision. ” to: “Inclusion criteria were that children have no reading and learning disabilities, attention disorders, have normal or corrected to normal vision according to teachers assessment.” (LINE 99)

Point 7:

- Page 3, Line 116: data real-time monitoring >> real-time data monitoring

Response 7: The sentence “Two laptops were used for data real-time monitoring and storage…” is changed to “Two laptops were used for real-time data monitoring and storage…” (LINE 121)

Point 8:

- Page 3: I don't seem to find any information about participants' baseline reading level and their reading experience (e.g., how many hours they typically read per week, etc.).

Response 8: We can only estimate the total reading hours per week. It would be about two hours per day and about 10 hours per week (counting working days only). However, and more importantly, the reading experience was compatible between the two groups, as both of these grades spend equal amounts of time in school (8am-3.20pm) and they finish all their homework at school as well.

We added the following sentence to the manuscript: “The reading experience was compatible between the two grade groups, as both of these grades spend equal amounts of time in school (8am-3.20pm) and they finish all their homework at school as well.” (LINE 94 to 96)

Point 9:

- Page 3, Lines 162-163: It is unclear what happened after the investigator(s) assessed comprehension. Did the investigators record scores or summative evaluation?

Response 9: We did not analyse the comprehension questions further. Those were just the basic questions (such as: who is the main character, where does this story take place, what did the man do etc...), not the ones where we would assess a deeper understanding of the text. It was more a control whether or not children were attentive to the text and they all knew answers to those basic questions.

We have rephased the sentence: “After finishing the test, the researcher checked with each child the level of understanding of the story with questions recommended in the literature after the story for exercise.” to the following: “After finishing the test, the researcher posed questions (recommended for exercise in the literature after the story) to check whether the children read the text carefully or not.” (LINE 167 to 169)

Point 10:

- Page 4, Line 187: BeGaze 3.7 software. Like it was done for Matlab in a later sentence, need to provide the company name, and location information. Or, if BeGaze is part of the eye tracking system, clearly indicate so.

Response 10: We have specified this by completing the sentence: “Eye tracking data analysis and visualization was performed using BeGaze 3.7 software.” to “Eye tracking data analysis and visualization was performed using SMI BeGaze TM 3.7 software (SMI, Germany).” (LINE 194 to 195)

Also we have changed the following sentence: “An SMI RED-m 120-Hz portable remote eye tracker (https://www.smivision.com) was placed below the computer monitor in front of the participant, and it was fixed in place to keep it from accidentally moving. ” to “An SMI RED-m 120-Hz portable remote eye tracker (SMI, Germany, https://www.smivision. com) was placed below the computer monitor in front of the participant, and it was fixed in place to keep it from accidentally moving. ” (LINE 128)

Point 11:

- Page 5, Lines 224-235: I am not sure why this is here. It sounds like the author's instruction from a specific journal.

Response 11: It was included by mistake/copied from the instruction for authors. We have removed the whole section:

Materials and Methods should be described with sufficient details to allow others to replicate and build on published results. Please note that publication of your manuscript implicates that you must make all materials, data, computer code, and protocols associated with the publication available to readers. Please disclose at the submission stage any restrictions on the availability of materials or information. New methods and protocols should be described in detail while well-established methods can be briefly described and appropriately cited.

Research manuscripts reporting large datasets that are deposited in a publicly available database should specify where the data have been deposited and provide the relevant accession numbers. If the accession numbers have not yet been obtained at the time of submission, please state that they will be provided during review. They must be provided prior to publication.

Interventionary studies involving animals or humans, and other studies require ethical approval must list the authority that provided approval and the corresponding ethical approval code.”  (LINE 231 to 242)

Point 12:

- Methods: given the number of outcome measures, please report how much they are correlated each other to help the reader understand how much distinct information each outcome measure offers.

Response 12: We have computed Pearson’s correlations across all the parameters and found that EEG measures are significantly correlated within themselves, same is true for eye-tracking and heart rate measures. However, regarding the cross-methodologies correlations, we only found a strong positive correlation between fixation and saccade frequencies and heart rate measures, as well as reading duration and eye-tracking measures in expected direction which was not surprising. Based on these results, we can conclude that these different measures are not redundant, and they are all valuable to fully understand the process of reading. However, we would rather not expand the original story of the paper in this direction, unless you, or the editor, insists on it. Also, we are sending you the matrix of correlation so that you can see the above described pattern of correlations:

https://docs.google.com/spreadsheets/d/1Da9pTAY0ohHgu0h0LVfT8cAVLbDUd-fI/edit?dls=true#gid=931701098

Point 13:

- Page 6, Section 2.4: Please indicate whether or not there was a substantial departure from parametric test assumptions for any of dependent variables. It is unclear.

Response 13: We did not find a substantial departure from parametric test assumptions for any of dependent variables.

Point 14:

- Page 6, Table 2: EEG parameters. Please include the unit (Hz?). And n = 12 for the both grades. Previously, it was noted that one participant's data was excluded.

Response 14: In the Table 2, 3 and 4 we added the unit (squared microvolts) for EEG parameters (median Alpha, Beta, Delta, Theta and Whole Range power band): mV2.

Point 15:

- Page 7, Table 3: I don't think there is much value in noting that p<0.05 and p<0.01. Nowadays it is more encouraged to report the effect size. So, I also encourage the authors to remove such a distinction based on p values, and include the effect size. And n = 156. It is unclear where that came from.

Response 15: We have computed Cohen’s D effect sizes, and reported it in the table where we found significant differences between second and third graders. However, we still kept the significance levels based on the p-value, as for many colleagues in the field it would be a more informative statistic.

In the table legend we stated, instead of “Differences between second and third graders on reading duration, EEG, Eye tracking, EDA and HRV parameters (normalized on white colour), (p<.05 is marked with “+”, and p<.01 is marked with “++”)” it now says:  “Differences between second and third graders on reading duration, EEG, Eye tracking, EDA and HRV parameters (normalized on white colour), (p<.05 is marked with pale orange colour, and p<.01 is marked with dark orange colour). The values in the table represent Cohen’s D effect sizes”. (LINE 273 to 274)

Thank you for spotting n=156 in Table 4. It has been corrected, so that we have for the second grade n=12 and for the third grade n=12, like in the Table 2 for white colour.

Point 16:

- Page 9, Figure 2: The figures are not easy to read (e.g., label font size is too small, image quality is low, etc.). In addition, the variable in the horizontal axis is not a continuous variable, so a bar graph is more proper. Or, remove the line that connects the values of an outcome measure.

Response 16: The Figure 2 has been updated according to your suggestions.

Point 17:

- Page 10, Line 297: In the methods, it was noted that one participant data was excluded.

Response 17: Yes, and we have specified that in the methods section, when we described participants: “Only one child was excluded from the analysis due to the large artefacts in the acquired signals and his data were not used in the statistical analysis.” (LINE 99 to 100)

Point 18:

- Discussion: Given the two hypotheses were provided, please more clearly indicate whether or not the hypotheses were accepted or failed to be accepted.

Response 18: We have clarified this in the discussion, so that the text which was: “The corresponding graphics across colours are presented in Figure 2, where it could be observed that blue and turquoise backgrounds have a calming impact (increasing normalized SDNN, CVRR, and STD HR values) on second graders, which is in accordance with the previous reports [37-39].” it now reads as “The corresponding graphics across colours are presented in Figure 2, where it could be observed that blue and turquoise backgrounds have a calming impact (increasing normalized SDNN, CVRR, and STD HR values) on second graders, which is in accordance with the previous reports [19-21] and partially in accordance with our first hypothesis.  (LINE 335 to 336)

Also, this section has been changed: “The findings concerning Reading duration on background and coloured overlays showed that the third-grade students have longer Reading duration in comparison to the second-graders. This could be explained by the fact that third graders or older children need more time to adapt to unexpected text and stimuli such as colour [40–42].” so that it now reads as: “The findings concerning Reading duration on background and coloured overlays showed that the third-grade students have longer Reading duration in comparison to the second-graders. This could be explained by the fact that third graders or older children need more time to adapt to unexpected text and stimuli such as colour [22–24] which goes in line with our alternative hypothesis.” (LINE 346 to 347)

Reviewer 3 Report

The experimental in this paper is well designed. The multimodal EEG/ECG/EDA system and eye-tracking system used in the study can quantitatively describe the concentration and tension of children in the reading process. And this paper can analyze the effect of color on reading with different color backgrounds. However, there are some questions that are not fully addressed, and publication at this stage feels a bit premature.

1. Is it too far-fetched, that the effect of text background color on children's reading will change at different age? Moreover, the research object are second and third students, which might be questionable that if such age variation of one year can really reflect sharp change under this influence?

2. It is unreasonable to randomly select participants from a single class, because the learning atmosphere and teaching mode in one class may preset the children's reading habits and abilities. Choosing participants from a school or a larger range will make the results more convincing.

3. Sensors are very bulky, and the uncomfortableness of wearing them needs to be improved. Because, in this experimental environment, children's reading ability might be affected by the sensors, and reading abilities shown in this situation may differ from their normal reading abilities in a normal environment.

4. How can the author distinguish the influence from other environmental factors. Take the paragraph complexity for example. The reading ability result may be affected a lot by the paragraph complexity which various from person to person. The author should prove more evidence about that.

5. Here, in the conclusion of this paper(line 361 to 364), the author has mentioned “second graders show lower scores of reading duration and eye-tracking measures (fixation duration total and fixation duration average), which confirms the (alternative) hypothesis that older children need more time to adapt on unexpected text and have longer fixations on words during reading.”. However, the average reading duration of third graders(43.38) is only about 11% higher than second graders(49.07), that such low variance have very weak convincingness or the author should provide more evidence.

Author Response

Point 1:

Is it too far-fetched, that the effect of text background color on children's reading will change at different age? Moreover, the research object are second and third students, which might be questionable that if such age variation of one year can really reflect sharp change under this influence?

Response 1:

Thank you for this question. These two age groups are selected intentionally, because, kids in Serbia start to learn letters in the first grade and the biggest shift in the reading happens between the second and third graders. We, in fact, tried to test a few kids in the first grade as well, in order to catch them as early as possible, but the task was too tiring for them. Also, when testing kids from subsequent age groups we found that they take the same time and approach as third graders. So, we believe that the selection is well justified by the education program, as well as through the additional testing which was not reported in the manuscript. But, we have now justified the grade selection in the manuscript as well:

“These two age groups were selected intentionally, because kids in Serbia start to learn letters in the first grade and the biggest shift in the reading happens between the second and third graders.“ (LINE 92-94)

Also, our pilot study showed that for the first grade students this task was too long and tiring, whereas, third year students are fully proficient in reading.

Point 2:

It is unreasonable to randomly select participants from a single class, because the learning atmosphere and teaching mode in one class may preset the children's reading habits and abilities. Choosing participants from a school or a larger range will make the results more convincing.

Response 2:

In fact, kids were selected from 8 different, random classes from a big school with over 900 pupils, not from a single class. We did not specify that clearly in the manuscript originally. But, we have now corrected that. Thank you. 

We have changed the following sentence in the manuscript: "Twenty-five healthy participants were randomly chosen from a single class of the second and third grades of the elementary school “Drinka Pavlović” in Belgrade respectively", so that it now reads as: "Twenty-five healthy participants were randomly chosen from eight classes of the second and third grades of the elementary school “Drinka Pavlović” in Belgrade, respectively.” (LINE 91)

 Point 3:

Sensors are very bulky, and the uncomfortableness of wearing them needs to be improved. Because, in this experimental environment, children's reading ability might be affected by the sensors, and reading abilities shown in this situation may differ from their normal reading abilities in a normal environment.

Response 3:

This was our concern before the start of the experiment as well, and for that reason we have chosen the most comfortable EEG system (mBrainTrain mobile version and remote eye-tracker), with soft sponges, which is very comfortable, rather than NeuroScan for example (which uses gel and syringes for application). Also, we can assure you that kids loved it, they were coming back to the experimenter to ask if they can take part in it again or if we needed further help.

 And, in a PISA study which we ran recently with a similar set-up, contrasting preselected excellent and poor students, both groups showed much better performance in this “cool” setting as they described it in comparison to the standard PISA test on which they were preselected. So, kids seem to be perceiving this situation as a scientific adventure which we encouraged.

In the present study, kids received a scientific diploma and a small present at the end of the testing session.

Point 4:

How can the author distinguish the influence from other environmental factors. Take the paragraph complexity for example. The reading ability result may be affected a lot by the paragraph complexity which various from person to person. The author should prove more evidence about that.

Response 4:

We have clarified this in the method section - we used pseudo-randomisations in order to avoid the possibility of any paragraph to be fixed to and read on a single colour. By pseudo-randomising the order of colours in relation to paragraphs, we hoped that any paragraph-related specificities and complexities would average out. Otherwise, we could only agree with you - it would be impossible to control and keep the paragraph complexity at the same level.

The paragraph in the manuscript was the following: "The experiment was run applying pseudo randomization of colour background/overlay order starting always with a referent slide (black text on white background). No other colour was fixed/related to a certain text. Therefore, in this way, any other factors apart from the actual colour would be averaged out (paragraph complexity such as vocabulary, syntax etc, as well as semantic/affective content)." (LINE 113 to 117)

Point 5:

Here, in the conclusion of this paper (line 361 to 364), the author has mentioned “second graders show lower scores of reading duration and eye-tracking measures (fixation duration total and fixation duration average), which confirms the (alternative) hypothesis that older children need more time to adapt on unexpected text and have longer fixations on words during reading.”. However, the average reading duration of third graders(43.38) is only about 11% higher than second graders(49.07), that such low variance have very weak convincingness or the author should provide more evidence.

Response 5:

We had to report this result as significant, although it was only 11%, because according to the statistical test it was a significant and reliable effect. Even if it was smaller, but significant, we would have to report and interpret it as such.

Round 2

Reviewer 2 Report

Thanks for the response.  I don't have a major concern, but have a few minor suggestions.

  • Line 118.  It is right now shown as a single sentence paragraph ("Details about ... Design.).  This can be placed in the previous paragraph or the follow-up paragraph, I think. 
  • Lines 196-197: Again, it is a single sentence paragraph. Can be placed in the previous paragraph.
  • In the statistical analysis section, the authors need to report that there were no substantial deviations from parametric testing assumptions.
  • Line 223: p -> p
  • In the results section, I think there is a small formatting issue with subsection numbering.  "White (default) background - reading results" should be 3.1, I think. 

Author Response

Point 1:

Line 118.  It is right now shown as a single sentence paragraph ("Details about ... Design.).  This can be placed in the previous paragraph or the follow-up paragraph, I think. 

Response 1: We have placed the sentence “Details about the “experiment” content are explained in the Section Experiment Design.” in the previous paragraph. (LINE 117 to 118)

Point 2:

Lines 196-197: Again, it is a single sentence paragraph. Can be placed in the previous paragraph.

Response 2: We have placed the sentence “EEG/ECG/EDA data were analysed using Matlab ver. 8.5 (Mathworks, USA) in the manner described below. in the previous paragraph. (LINE 196 to 197)

Point 3:

In the statistical analysis section, the authors need to report that there were no substantial deviations from parametric testing assumptions.

Response 3: We added the sentence “There were no substantial deviations from parametric testing assumptions.” in the Section 2.4. Statistical Methodology (LINE 224 to 225).

Point 4:

Line 223: p -> p

Response 4: We have changed “p” to italic: “All p values…” to “All p values…”. (LINE 223)

Point 5:

In the results section, I think there is a small formatting issue with subsection numbering.  "White (default) background - reading results" should be 3.1, I think. 

Response 5: Thank you for this observation. We have corrected the section numbering: 3.1 is now the section number for “White (default) background - reading results” subtitle and 3.2 is now the section number for “Background and overlay colours – reading results” subtitle. (LINE 232 and LINE 245, respectively)

Reviewer 3 Report

Accept in present form

Author Response

Thank you!

Kind regards,

Tamara Jakovljević et al.